# Intelligent Identification Method of Geographic Origin for Chinese Wolfberries Based on Color Space Transformation and Texture Morphological Features

**DOI:** 10.3390/foods12132541

**Published:** 2023-06-29

**Authors:** Jiawang He, Tianshu Wang, Hui Yan, Sheng Guo, Kongfa Hu, Xichen Yang, Chenlu Ma, Jinao Duan

**Affiliations:** 1College of Artificial Intelligence and Information Technology, Nanjing University of Chinese Medicine, Nanjing 210023, China; daaang@126.com (J.H.); kfhu@njucm.edu.cn (K.H.); chenluma1217@126.com (C.M.); 2National and Local Collaborative Engineering Center of Chinese Medicinal Resources Industrialization and Formulae Innovative Medicine, Jiangsu Collaborative Innovation Center of Chinese Medicinal Resources Industrialization, Nanjing University of Chinese Medicine, Nanjing 210023, Chinadja@njucm.edu.cn (J.D.); 3School of Computer and Electronic Information and School of Artificial Intelligence, Nanjing Normal University, Nanjing 210023, China; xichen_yang@njnu.edu.cn

**Keywords:** Chinese wolfberry, geographic origin identification, image processing, Hu invariant moment, Gabor transform, random forest

## Abstract

Geographic origins play a vital role in traditional Chinese medicinal materials. Using the geo-authentic crude drug can improve the curative effect. The main producing areas of Chinese wolfberry are Ningxia, Gansu, Qinghai, and so on. The geographic origin of Chinese wolfberry can affect its texture, shape, color, smell, nutrients, etc. However, the traditional method for identifying the geographic origin of Chinese wolfberries is still based on human eyes. To efficiently identify Chinese wolfberries from different origins, this paper presents an intelligent identification method for Chinese wolfberries based on color space transformation and texture morphological features. The first step is to prepare the Chinese wolfberry samples and collect the image data. Then the images are preprocessed, and the texture and morphology features of single wolfberry images are extracted. Finally, the random forest algorithm is employed to establish a model of the geographic origin of Chinese wolfberries. The proposed method can accurately predict the origin information of a single wolfberry image and has the advantages of low cost, fast recognition speed, high recognition accuracy, and no damage to the sample.

## 1. Introduction

Chinese wolfberry is the dry and mature fruit of *Lycium barbarum* L., which is a plant of the Solanaceae family. Chinese wolfberry has the effects of nourishing the liver and kidneys and improving eyesight [1]. From 2011 to 2020, Chinese wolfberry consumption increased by an average of 9.7%. From 2012 to 2022, the Chinese national wolfberry planting scale increased rapidly, and the total output increased by more than two times [2]. At present, the Chinese wolfberry market has a large demand and a wide planting area. However, Chinese wolfberries from different origins are different in the growth environment, cultivated varieties, harvest period, processing, and storage, which leads to differences in the use of clinical compatibility [3]. This shows that studying the geo-herbalism of Chinese wolfberry has great significance for clinical treatment.

Nowadays, the methods for identifying the origin of food or medicinal materials mainly include artificial naked eye recognition [4], chemical composition analysis [5,6,7,8,9], near-infrared spectral analysis [10,11,12], image recognition, and other methods [13,14,15]. The traditional artificial naked eye method has low identification efficiency, and the screening results are affected by subjective factors. At present, there are many studies on the determination of the chemical composition of food or medicinal materials as identification features. Zinicovscai et al. have predicted the origin of wine by measuring trace elements in wine [5]. Takashima et al. have predicted the origin of the product by determining the DNA of several types of aquatic products [6]. Zhao et al. have analyzed the chemical components of Chinese wolfberry from different origins, and the experimental results showed that the overall characteristics of water-soluble nutrients in the samples were quite different [7]. Liu et al. proposed that there are significant differences in the polysaccharide, total flavonoids, and total phenolic contents of wolfberry leaves in different planting areas of wolfberry [8]. Bai et al. have established the HPLC (High Performance Liquid Chromatography) fingerprint of wolfberry, identified the main common peaks by LC-MS (Liquid Chromatography with Mass Spectrometry), and performed cluster analysis and discriminant analysis on the samples of wolfberry from four main producing areas [9]. However, these methods require trained technicians to extract the components of the sample through professional means and instruments. The process is complicated to operate and requires expensive instrumentation, so it is not universally popular.

In academia, there is also a method of collecting near-infrared spectra of food or medicine as an identification feature. Zhu et al. have used near-infrared spectroscopy to predict and discriminate the types of tobacco origin [10]. Tang et al. have used near-infrared spectroscopy to scan forty samples of wolfberry from eight different origins [11]. Wang et al. have used near-infrared hyperspectral images to identify the origin of wolfberry from five sources [12]. However, the collection of near-infrared spectra requires grinding wolfberry into powder, resulting in the loss of experimental materials. Moreover, the near-infrared spectrometer is expensive, and it is difficult for small enterprises to afford the instrument’s cost, so this method also has certain limitations.

It can be seen that whether it is to extract the chemical components of food or medicinal materials or to collect near-infrared spectra as identification features, there are limitations, such as high cost, strong professionalism, and low identification efficiency. Machine learning is a branch of artificial intelligence that enables computers to automatically learn the laws of data through models and algorithms, so as to realize the prediction and classification of new data. There are mainly supervised, unsupervised, semi-supervised, and reinforcement learning types, among which supervised learning is the most common. Algorithms are very important in machine learning, such as linear regression, decision trees, neural networks, etc., and feature engineering is also an important link. Machine learning technology is widely used in natural language processing, image recognition, medical diagnosis, and other fields. With the development of computer vision and machine learning technology, many researchers have established an image-based origin identification model to identify the origin of food or medicinal materials by collecting images of samples and conducting training and learning. De et al. have identified charcoal sources using macro images and deep learning algorithms [13]. Wang et al. have used the methods of image and visual information and machine learning to intelligently identify the origin of Angelica [14]. Wang et al. have presented an efficient and convenient identification method based on image processing [15]. The existing research can fully show that the method of origin identification based on images has the advantages of non-destructive samples, low cost, a high recognition rate, and fast recognition speed.

It has been proposed that texture morphology is an important feature for identifying objects or regions of interest in any image [16]. Studies have shown that Chinese wolfberries produced from different origins are significantly different in shape, size, and texture. For example, the texture of the Chinese wolfberry from Qinghai province is relatively vague, and the size is medium. The texture of the Chinese wolfberry from Qinghai province is particularly clear, and its size is relatively large [4]. Therefore, it is feasible to extract the texture and morphological features in the image to identify the geographic origin of Chinese wolfberry. The HIS (Hue Saturation Intensity) color space is based on the principle of human eye imaging and presents the hue, saturation, and brightness information of the image [17,18]. Converting the original image into HSI space can weaken the influence of light changes, thereby enhancing the stability of the algorithm. Based on HSI, the color image of standing trees in the forest area is segmented and accurately extracted from the background [19]. In this paper, after converting the original image into the HSI color space, the Gabor transformation is separately performed on the H channel, S channel, and I channel to extract the texture features of the Chinese wolfberry. Finally, the Hu invariant moments are calculated to extract the morphological features. The texture features of the image can be well extracted through the transformation of Gabor at different angles [20]. The Hu invariant moment feature has the characteristics of translation, rotation, and scale invariance [21], and the morphological characteristics of wolfberries can be preserved by calculating the Hu invariant moment. The main contributions of this study are as follows:

The established wolfberry image database contains images of the Chinese wolfberry from the four major production areas: Gansu, Inner Mongolia, Ningxia, and Qinghai. The images have been marked with labels of origin.The shape and texture features of the Chinese wolfberry are represented. The features can accurately reflect the characteristics of the Chinese wolfberry.An intelligent identification method for the geographic origin of the Chinese wolfberry is presented, and the accuracy of this method has increased by more than 60% compared with the existing related methods.

## 2. Methods

In this paper, the identification method of geographic origin for Chinese wolfberry based on color space transformation and texture morphological features is proposed. The flow chart is shown in Figure 1. First of all, the data on wolfberry samples was collected by a high-definition digital camera. Secondly, the image is cropped to a suitable size, and the background of the image is processed into a uniform black. Furthermore, after transforming the image into HSI space, the texture and shape of the image are extracted as recognition features. Finally, the random forest (RF) algorithm is employed to establish the identification model of the geographic origin of Chinese wolfberries.

### 2.1. Sample Preparation and Image Acquisition

In China, Ningxia, Gansu, and Qinghai are the main producing areas of the Chinese wolfberry [22]. A total of 90 single wolfberries were collected from four producing areas, i.e., Ningxia, Gansu, Qinghai, and Inner Mongolia, as experimental samples and image data. The images were captured with a Canon full-frame digital camera, the EOS 5DS R, and a Sigma wide-angle lens of 24–35 mm under the same conditions to ensure the consistency and integrity of the texture and shape of the Chinese wolfberries. The original information is shown in Table 1.

In Table 1, GS means Gansu, NM means Inner Mongolia, NX means Ningxia, and QH means Qinghai. The sample images of the Chinese wolfberries from the four origins are shown in Figure 2. The pixels of the four pictures are 8688 × 5792. From Figure 2, it is difficult for us to judge the origin type of the Chinese wolfberry with the naked eye.

### 2.2. Image Preprocessing

The use of a wide-angle lens makes the size of a single wolfberry too small. Therefore, the captured wolfberry images have a large amount of blank space, which will have a significant impact on the training and recognition of the model. In this paper, the image is preprocessed to remove invalid background information. First, the RGB images are converted into grayscale images by Formula (1). In the formula, *R_ij_*, *G_ij_*, and *B_ij_* are the values of the pixel points in row *i* and column *j* of the R channel, G channel, and B channel in the original image of wolfberry, respectively. *Gray_ij_* represents the grayscale information of the pixel in row *i* and column *j* of the image.
(1)$$Gray_{ij}=0.299*R_{ij}+0.578*G_{ij}+0.114*B_{ij}$$

Then the OTSU (Maximum Between-Class Variance) algorithm is used to select a specific threshold to minimize the intra-class variance of the thresholded black and white pixels. Let $\left\{0,1,2,\ldots ,L-1\right\}$ represent the different gray levels in the digital image, and $n_{i}$ represents the number of pixels with the gray level. Then the threshold is set as k,(0<k<L−1). The maximum separable measure $\eta_{max}$ can be calculated by the Formulas (2)–(6), where n represents the total number of pixels in the image and satisfies $n=n_{0}+n_{1}+\cdots +n_{L-1}$. $p_{i}$ represents the probability that the gray level of the pixel is i and satisfies $p_{i}=\frac{n_{i}}{n}$. $P_{1}\left(k\right)$ represents the probability that the pixel is classified into the first class and satisfies $P_{1}\left(k\right)=\sum_{i=0}^{k}p_{i}$. $P_{2}\left(k\right)$ represents the probability that the pixel is classified into the second class, satisfies $P_{2}\left(k\right)=\sum_{i=k+1}^{L-1}p_{i}$. $m_{1}\left(k\right)$ indicates the average gray value of pixels classified in class 1 and $m_{2}\left(k\right)$ represents the average gray value of pixels classified in class 2. $m_{G}$ indicates the average gray value of the entire image, which satisfies $m_{G}=\sum_{i=0}^{L-1}ip_{i}$. $\sigma_{B}^{2}$ represents the between-class variance and $\sigma_{G}^{2}$ represents the global variance.
(2)$$m_{1}(k)=\sum_{i=0}^{k}iP(i|C_{1})=\frac{1}{P_{1}(k)}\sum_{i=0}^{k}ip_{i}$$
(3)$$m_{2}(k)=\sum_{i=k+1}^{L-1}iP(i|C_{2})=\frac{1}{P_{2}(k)}\sum_{i=k+1}^{L-1}ip_{i}$$
(4)$$\sigma_{B}^{2}=P_{1}{\left(m_{1}-m_{G}\right)}^{2}+P_{2}{\left(m_{2}-m_{G}\right)}^{2}=P_{1}P_{2}(m_{1}-m_{2})$$
(5)$$\sigma_{G}^{2}={\sum_{i=0}^{L-1}\left(i-m_{G}\right)}^{2}p_{i}$$
(6)$$\eta_{\max}=\max_{0\leq k\leq L-1}\frac{\sigma_{B}^{2}(k)}{\sigma_{G}^{2}}$$

Next, according to the calculated maximum separable measure $\eta_{\max}$, set the gray value of the pixel point greater than or equal $\eta_{\max}$ to 1, and set the gray value of the pixel point smaller than $\eta_{\max}$ to 0. The obtained image BW is shown in Figure 3a. Then the Sobel edge detection algorithm is used to extract the contour information of the two-dimensional image. Let $g_{x}$ denote the horizontal Sobel convolution factor and $g_{y}$ denote the vertical Sobel convolution factor. $G_{x}$ indicates the image grayscale value of horizontal edge detection, satisfies $G_{x}=g_{x}\times BW$, $G_{y}$ indicates the image grayscale value of vertical edge detection, satisfies $G_{y}=g_{y}\times BW$. According to the Formulas (7)–(9), the new image G can be obtained as shown in Figure 3b.
(7)$$g_{x}=\left[\begin{array}{lll} -1 & 0 & 1\\ -2 & 0 & 2\\ -1 & 0 & 1 \end{array}\right]$$
(8)$$g_{y}=\left[\begin{array}{ccc} 1 & 2 & 1\\ 0 & 0 & 0\\ -1 & -2 & -1 \end{array}\right]$$
(9)$$G=\sqrt{G_{x}^{2}+G_{y}^{2}}$$

The non-zero abscissa of the image G is denoted as x, and $x_{i}$ represents the ith abscissa. Then $x_{i+1}-x_{i}$ is calculated; if the result is greater than 10, it means that there is noise. Then set $xi_{\min}$ in the image at this time. In the same way, $x_{\max}$, $y_{\min}$, $y_{\max}$ can be obtained. Finally, arrange the above four values to obtain four endpoints. The cropped area is a rectangular area surrounded by four points. Figure 3c shows the cropped effect diagram; compared with Figure 3a, the redundant blank area is obviously deleted. After performing morphological expansion and erosion operations on Figure 3c, the white background of the wolfberry image is deducted, and the image of Chinese wolfberry with a unified background is obtained as shown in Figure 3d.

During the entire image preprocessing process, the above algorithms are used to complete image cropping and background color unification. The purpose of this procedure is to reduce the interference of the background in subsequent model training. Figure 4 illustrates the preprocessed images of Chinese wolfberries from four origins. The wolfberries in Figure 4a–d are from Gansu, Inner Mongolia, Ningxia, and Qinghai, respectively. The resolution of the pixels for the four images is 886 × 947, 853 × 573, 514 × 1249, and 533 × 1024, respectively.

### 2.3. Gabor Transform Feature Extraction

The three-dimensional matrix of the HSI image (*H_ij_*, *S_ij_*, and *I_ij_*) is obtained according to the Formulas (10)–(13). Among them, $H_{ij}$ represents the hue of the pixel in row *i* and column *j*, $\theta_{ij}$ represents the hue angle of the pixel in row *i* and column *j*, $S_{ij}$ represents the saturation of the pixel in row *i* and column *j*, $I_{ij}$ represents the brightness of the pixel in row *i* and column *j*.
(10)$$H_{ij}=\left\{\begin{array}{l} 2\pi -\theta_{ij}\;\;G_{ij}<B_{ij}\\ \theta_{ij}\;\;G_{ij}\geq B_{ij} \end{array}\right.$$
(11)$$\theta_{ij}=\cos^{-1}\theta \left\{\frac{0.5\times [(R_{ij}-G_{ij})+(R_{ij}-B_{ij})]}{\sqrt{{\left(R_{ij}-G_{ij}\right)}^{2}+(R_{ij}-B_{ij})(G_{ij}-B_{ij})}}\right\}$$
(12)$$S_{ij}=1-\frac{3\times \min\left\{R_{ij},G_{ij},B_{ij}\right\}}{R_{ij}+G_{ij}+B_{ij}}$$
(13)$$I=\frac{R_{ij}+G_{ij}+B_{ij}}{3}$$

Then the HSI three-channel image information of the Chinese wolfberry can be obtained, as shown in Figure 5. Figure 5a illustrates a three-channel overlay image. Figure 5b is an H channel image; Figure 5c is an S channel image; and Figure 5d is the I channel image.

Following that, the Gabor transform is performed on the three channels of the HSI image, respectively. Firstly, a two-dimensional Gabor filter function is generated according to the Formulas (14)–(16), where x and y represent the pixel coordinates, $x_{p}$ and $y_{p}$ represent the coordinate transformation variables, respectively, and λ represents the wavelength of the sinusoidal component. The wavelength controls the width of the Gabor function strips. θ controls the direction of the Gabor function, with zero degrees corresponding to the vertical position of the Gabor function. γ controls the aspect ratio, or height of the Gabor function. σ controls the bandwidth, or the overall size of the Gabor envelope. φ indicates the relative offset, which is the relative offset of the tuning function.
(14)$$G(x,y,\lambda ,\theta ,\varphi ,\sigma ,\gamma )=\exp(-\frac{x_{p}^{2}+\gamma^{2}y_{p}^{2}}{2\sigma^{2}})\exp(i(2\pi \frac{x_{p}}{\lambda }+\varphi ))$$
(15)$$x_{p}=x*\cos\theta +y*\sin\theta$$
(16)$$y_{p}=y*\cos\theta -x*\sin\theta$$

Finally, the image is convolved with the real part and the imaginary part of the Gabor filter function. The size of the convolution result is consistent with the original image matrix. The real part matrix and the imaginary part matrix are fused to obtain the filtered matrix. Then the histogram of the image for each channel is calculated. Finally, the 1 × 256-dimensional features obtained from the three channels are formed to get a 1 × 768 texture feature matrix.

The textures of Chinese wolfberries from different origins are varied [4]. Gabor transformation can extract the texture features in wolfberry images [20]. Figure 6 illustrates the images after Gabor transformation in the I channel. From Figure 6, the texture of the wolfberry samples from Inner Mongolia and Qinghai is more rugged, with large ravine stripes. While the wolfberry samples from Gansu and Ningxia are more delicate. Therefore, the histogram is calculated after Gabor transformation to reflect the distribution of different texture characteristics for the wolfberries from different origins.

### 2.4. Hu Invariant Moment Feature Extraction

The Hu invariant moment feature has the characteristics of translation, rotation, and scale invariance [19], which can well preserve the morphological characteristics of wolfberries. First, the RGB images are converted into grayscale images by Formula (1). Let the size of the image matrix be m × n, and then calculate the (*p* + *q*) ordinary moment $m_{pq}$ and center distance $\mu_{pq}$ of the grayscale image according to the Formulas (17)–(19). Although the central moment has translation invariance, it still does not have scale invariance, so the center distance must be normalized to obtain $e_{pq}$ by the Formula (20). Finally, the calculation of the Hu invariant moment is completed by Formulas (21)–(27), thus obtaining a 7-dimensional feature: {H1, H2, H3, H4, H5, H6, H7}. Among them, $\overset{¯}{x}$ and $\overset{¯}{y}$ indicate the centroid coordinates of the target area.
(17)$$m_{pq}=\sum_{x=1}^{m}\sum_{y=1}^{n}x^{p}y^{q}*Gary_{xy}\;\;p,q=0,1,2,\cdots$$
(18)$$\bar{x}=\frac{m_{10}}{m_{00}},\bar{y}=\frac{m_{01}}{m_{00}}$$
(19)$$\mu_{pq}=\sum_{x=1}^{m}\sum_{y=1}^{n}{\left(x-\bar{x}\right)}^{p}{\left(y-\bar{y}\right)}^{q}*Gary_{xy}\;\;p,q=0,1,2,\cdots$$
(20)$$e_{pq}=\frac{\mu_{pq}}{\mu_{00}^{\frac{p+q+2}{2}}}\;\;p+q=2,3,\cdots$$
(21)$$H_{1}=e_{20}+e_{02}$$
(22)$$H_{2}={\left(e_{20}-e_{02}\right)}^{2}+4e_{11}^{2}$$
(23)$$H_{3}={\left(e_{30}-3e_{12}\right)}^{2}+{\left(3e_{21}-e_{03}\right)}^{2}$$
(24)$$H_{4}={\left(e_{30}+e_{12}\right)}^{2}+{\left(e_{21}+e_{03}\right)}^{2}$$
(25)$$\begin{array}{ll} H_{5} & =(e_{30}-3e_{12})(e_{30}+e_{12})[{\left(e_{30}+e_{12}\right)}^{2}-3{\left(e_{21}+e_{03}\right)}^{2}]\ldots \\ & +(3e_{21}-e_{03})(e_{21}+e_{03})[3{\left(e_{30}+e_{12}\right)}^{2}-{\left(e_{21}+e_{03}\right)}^{2}] \end{array}$$
(26)$$H_{6}=(e_{20}-e_{02})[{\left(e_{30}+e_{12}\right)}^{2}-{\left(e_{21}+e_{03}\right)}^{2}]+4e_{11}(e_{30}+e_{12})(e_{21}+e_{03})$$
(27)$$\begin{array}{ll} H_{7} & =(3e_{21}-e_{03})(e_{30}+e_{12})[{\left(e_{30}+e_{12}\right)}^{2}-3{\left(e_{21}+e_{03}\right)}^{2}]\ldots \\ & -(e_{30}-3e_{12})(e_{21}+e_{03})[3{\left(e_{30}+e_{12}\right)}^{2}-{\left(e_{21}+e_{03}\right)}^{2}] \end{array}$$

According to the above formulas, the Hu invariant moments of the wolfberry images are calculated. Figure 7 shows the Hu invariant moments of the wolfberry images from the four origins. It can be seen that there are differences in the Hu invariant moment characteristics of the four origins.

### 2.5. Model Training

In this paper, the 1 × 7-dimensional Hu invariant moment feature is fused with the 1 × 768-dimensional feature after Gabor transformation to form a 1 × 775-dimensional feature matrix $F^{tr}=\left\{f^{tr\_1},f^{tr\_2}\ldots f^{tr\_775}\right\}$. Since a total of 360 sample features were obtained from the four origins, a 360 × 775-dimensional data set was formed after all the features were fused. Let be S the sample space; let trainX be the features of the training set; let trainY be the label of the training set; let testX be the feature of the test set; and let testY be the label of the test set. The relationship between trainX and testX is shown in the Formulas (28)–(30).
(28)trainX∩testX=Ø
(29)trainX∪testX=S
(30)trainX:testX=8:2

The random forest algorithm is an integrated learning algorithm based on decision trees that performs classification and regression by constructing multiple decision trees. Each decision tree in a random forest is made by randomly selecting samples and features, so it has good generalization ability. According to the Formula (31), the RF algorithm was used to train the sample characteristics of the four origins of the Chinese wolfberry, and many decision trees were constructed to form the identification model.
(31)RF_MODEL=RF_train(trainX,trainY)

Then input testX into the RF random forest model according to the Formula (32) and get a series of predicted values denoted as PL.
(32)PL=RF_MODEL(testX)

Finally, according to the Formulas (33)–(35), the accuracy rate ACC of the model is obtained by comparing with testY and PL, where m is the total number of samples in the test set and n is the number of correct recognitions.
(33)ACC=(n/m)×100%
(34)$$n=\sum_{i=1}^{m}c_{i}$$
(35)$$c_{i}=\left\{\begin{array}{c} 1\;PL_{i}=testY_{i}\\ 0\;PL_{i}\neq testY_{i} \end{array}\right.$$

## 3. Experimental Results

In this experiment, MATLAB 2022a is used to construct the algorithm model. The CPU is AMD Ryzen 54,600 U, the memory is DDR4 3200 MHz 16 GB memory, and the operating system is Windows 10. The implementation of the RF algorithm uses the Classification Random Forest package [23]. The Gabor transformation function parameter settings are shown in Table 2.

In this paper, a total of 360 images of Chinese wolfberry samples are used for experiments. There are four origins, namely Gansu, Inner Mongolia, Ningxia, and Qinghai, and each origin has 90 sample images.

### 3.1. Determination of the Hyperparameters of RF

In order to determine the optimized hyperparameters of RF, five groups of comparative experiments were executed. The ratio of the training set to the testing set in each experiment is 8:2. The hyperparameter settings of the RF model are shown in Table 3. It can be concluded that when nTree is set to 2000 and mtry is set to 50, the model works best. Thus, in the prediction model, nTree and mtry are set to 2000 and 50, respectively.

### 3.2. Sensitivity to Training Set Size

In the proposed method, the training set samples are trained to obtain the prediction model. Thus, the training set size can affect prediction accuracy. In order to analyze the stability of the proposed method, nine groups of comparative experiments were set up. Each experiment uses the following proportions of the training set: 10%, 20%, 30%, 40%, 50%, 60%, 70%, 80%, and 90%. The train-test procedure is repeated 200 times. Figure 8 shows the average accuracy of the model for different training set ratios. It can be seen from Figure 8 that the model still has a high accuracy rate when the proportion of the training set is very low. This shows that the model proposed in this paper has high robustness.

### 3.3. Misjudgment Analysis

In this paper, the proportion of the training set is set to 80%, and the experiment is repeated 200 times. Then the number of errors, the number of selected samples, and the recognition error rate of each sample can be obtained. The error rates of samples from different origins are shown in Figure 9. The average error rate of samples from Inner Mongolia was the minimum of 5.51%, and the average error rate of samples from Qinghai was the maximum of 21.34%.

According to the analysis of the experimental data, 86% of the samples have a correct recognition rate of 100%, but 5% of the samples have a correct recognition rate of 0%. There are a total of 27 samples with an error rate higher than 90%, and the distribution of each origin is shown in Figure 10.

Among the samples with an error rate higher than 90%, there are 13 samples belonging to Qinghai, 5 samples belonging to Gansu, 4 samples belonging to Inner Mongolia, and 5 samples belonging to Ningxia. The specific error conditions can be seen in Table 4 below. As can be seen from Table 4, Qinghai wolfberry is more easily confused with Ningxia wolfberry. Inner Mongolian wolfberry and Ningxia wolfberry are easily predicted as Qinghai wolfberry, and Gansu wolfberry is easily predicted as Ningxia wolfberry.

Figure 11 illustrates the heat map of the confusion matrix obtained from one of the training results. It can be seen that while the model proposed in this paper has a high accuracy rate, Ningxia wolfberry is easily misjudged as Qinghai wolfberry.

### 3.4. Performance Comparison

The proposed method is compared with six other algorithms: CNN [4], KFDA [24], SVM [25], BPNN [26], Gabor_SVM, and Gabor_BPNN. CNN (Convolutional Neural Network) is a representative image recognition algorithm that has been widely used in the field of image recognition in recent years. In the CNN [4] algorithm, the epoch is set to 2000, the size of the batch is set to 64, and the learning rate is set to 0.001. In KFDA [24], the Gaussian kernel function is adopted, and the kernel function uses a matrix similarity measurement method based on Euclidean distance to determine the optimal kernel function. In Gabor_SVM and Gabor_BPNN, the Gabor transformation is used to take its mean, contrast, and entropy as the second feature to train the model. The loss function of SVM is set to 100, and the gamma function in the kernel function is set to 0.1. In BPNN (Back Propagation Neural Network), the maximum number of iterations is set to 1000, the target error of neural network training is 0.00001, and the learning rate is 0.01.

The images are divided into a training set and a test set with a ratio of 8:2. The number of images in the training set is 288, and the number of images in the test set is 72. Each algorithm is repeated 200 times, and the average accuracy is shown in Figure 12. It can be seen from the figure that the proposed methods, Gabor_SVM and Gabor_BPNN, have higher average accuracy than others. It indicates that the Gabor feature can effectively represent Chinese wolfberries of different origins. At the same time, the average accuracy of the proposed method is greater than the other six methods. Thus, the method proposed in this paper has unique advantages in the identification of the origin of Chinese wolfberry.

Among the proposed methods, Gabor_SVM and Gabor_BPNN, the error rate for the Qinghai region is always the highest. Figure 13 illustrates the proportion of Qinghai samples among the images with an error rate of more than 70%. It can be seen that the method proposed in this paper effectively reduces the recognition error rate of Qinghai samples.

## 4. Conclusions

Chinese wolfberry has a large planting area in China. It is very popular and has high nutritional value. The origin of wolfberry has a major impact on its medicinal value. This paper proposes an intelligent identification method for the geographic origin of the Chinese wolfberry based on color space transformation and texture morphological features that can quickly and efficiently identify the origin category of a single wolfberry image. The Chinese wolfberry samples are first prepared, and their image data are collected. The images are then preprocessed, and the Hu invariant moment and Gabor transformation features of the single wolfberry image are extracted. The RF algorithm is finally used to establish the identification model of the Chinese wolfberry’s origin. The experimental results show that, compared with other recognition algorithms, the proposed model has higher recognition accuracy and a better recognition effect.

## Figures and Tables

**Figure 1 foods-12-02541-f001:**
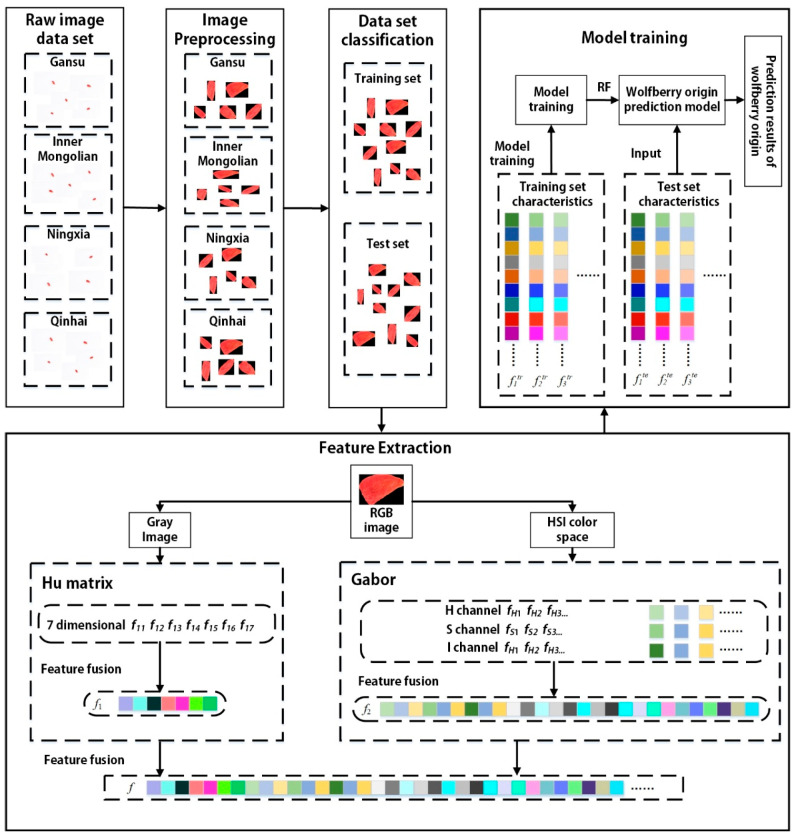
Flow chart of the proposed method.

**Figure 2 foods-12-02541-f002:**
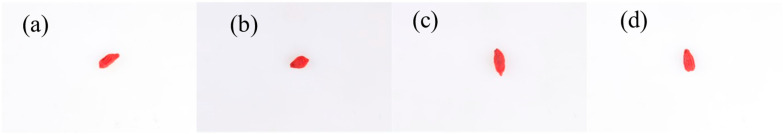
Image of wolfberry samples from four origin transformations: (**a**) Chinese wolfberry from Gansu Province; (**b**) Chinese wolfberry from Inner Mongolia; (**c**) Chinese wolfberry from Ningxia; (**d**) Chinese wolfberry from Qinghai.

**Figure 3 foods-12-02541-f003:**
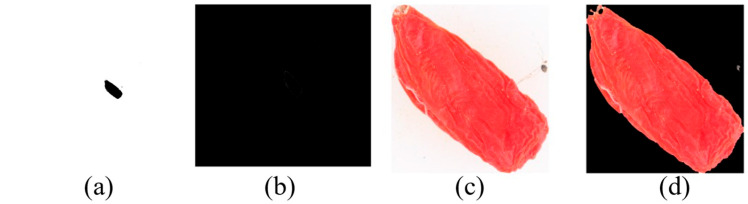
(**a**) Image processed by the OTSU algorithm; (**b**) Image processed by the Sobel algorithm; (**c**) Image after unifying the background; (**d**) Cropping effect after unifying the background.

**Figure 4 foods-12-02541-f004:**
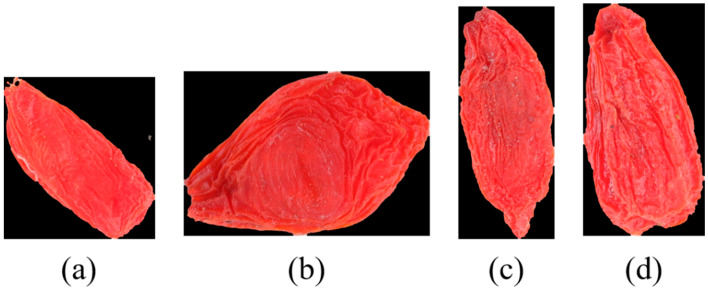
(**a**) Four origin images after preprocessing; (**a**) Chinese wolfberry from Gansu Province; (**b**) Chinese wolfberry from Inner Mongolia; (**c**) Chinese wolfberry from Ningxia; (**d**) Chinese wolfberry from Qinghai.

**Figure 5 foods-12-02541-f005:**
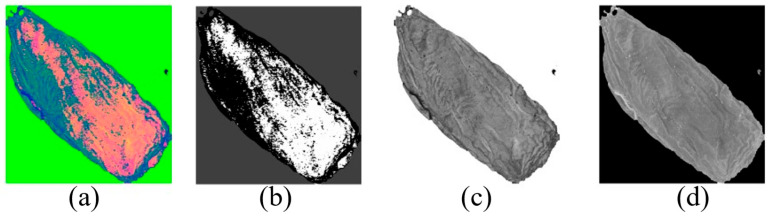
HSI three-channel information on the Chinese wolfberry; (**a**) Three-channel fusion image; (**b**) H-channel image; (**c**) S-channel image; (**d**) I-channel image.

**Figure 6 foods-12-02541-f006:**
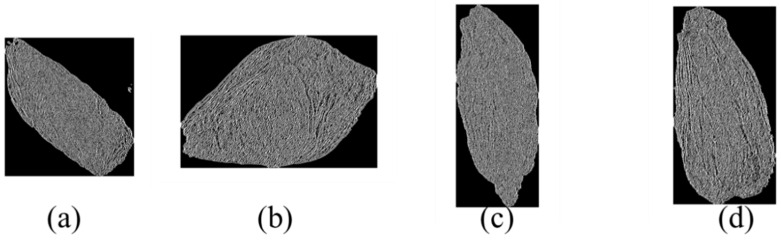
Texture information image after the I channel Gabor transformation. (**a**) Chinese wolfberry from Gansu Province; (**b**) Chinese wolfberry from Inner Mongolia; (**c**) Chinese wolfberry from Ningxia; (**d**) Chinese wolfberry from Qinghai.

**Figure 7 foods-12-02541-f007:**
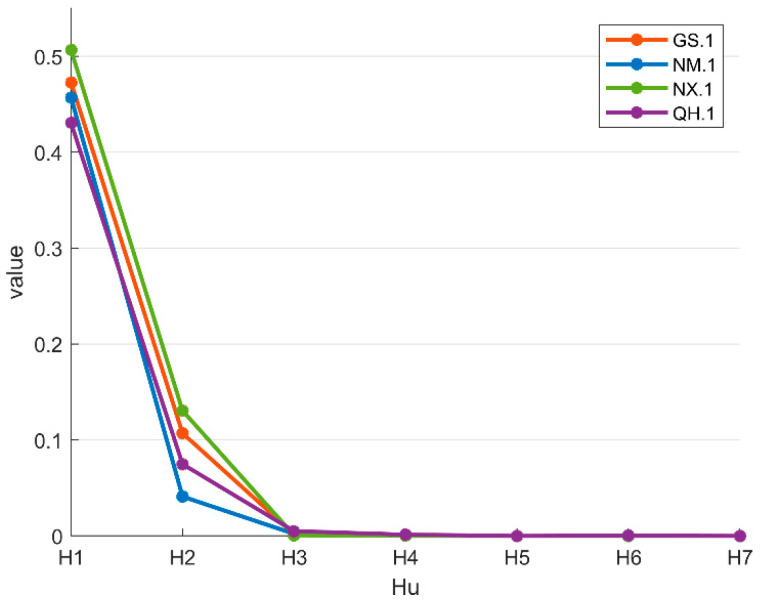
Hu invariant moment information of the wolfberry images from different origins.

**Figure 8 foods-12-02541-f008:**
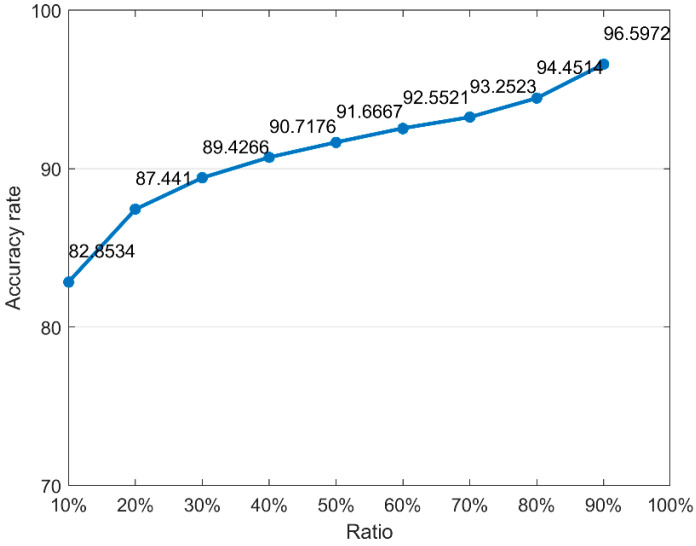
Training accuracy of different training set proportions.

**Figure 9 foods-12-02541-f009:**
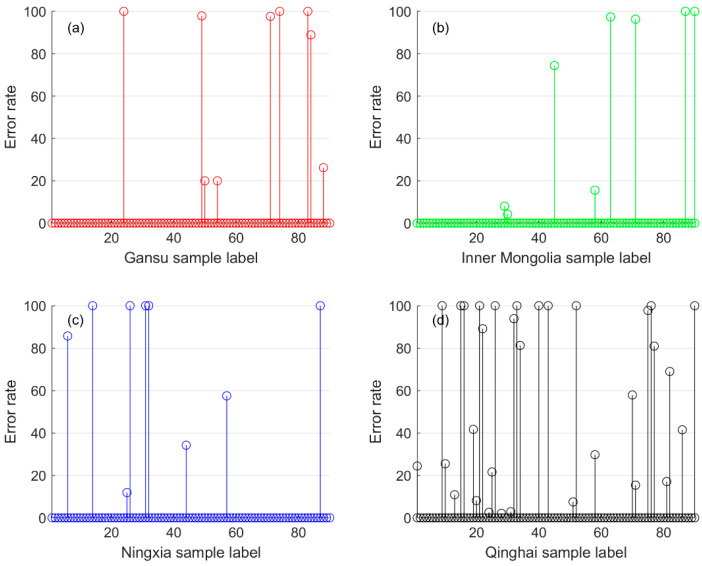
The relationship between the number of draws and the number of recognition errors. (**a**) Chinese wolfberry from Gansu Province; (**b**) Chinese wolfberry from Inner Mongolia; (**c**) Chinese wolfberry from Ningxia; (**d**) Chinese wolfberry from Qinghai.

**Figure 10 foods-12-02541-f010:**
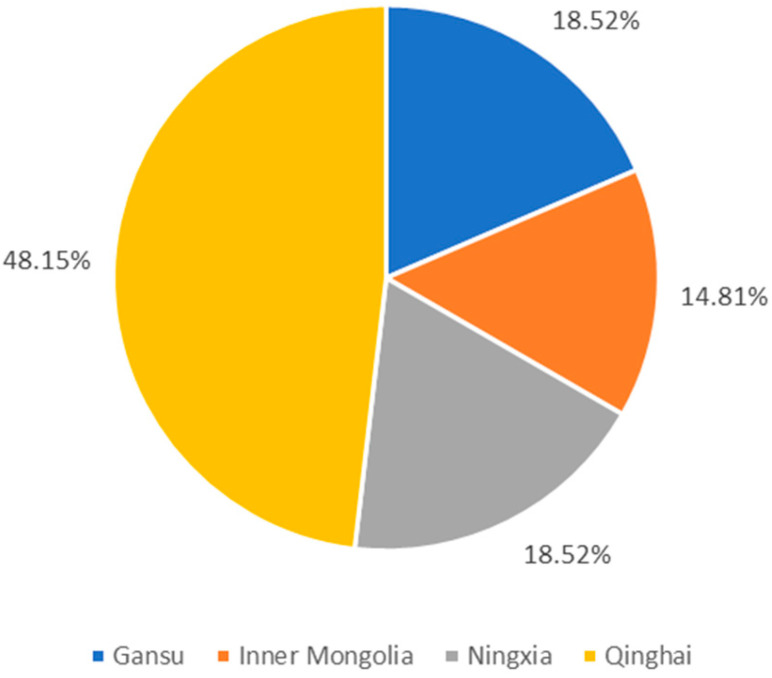
The distribution map of each origin in the samples with an error rate higher than 90%.

**Figure 11 foods-12-02541-f011:**
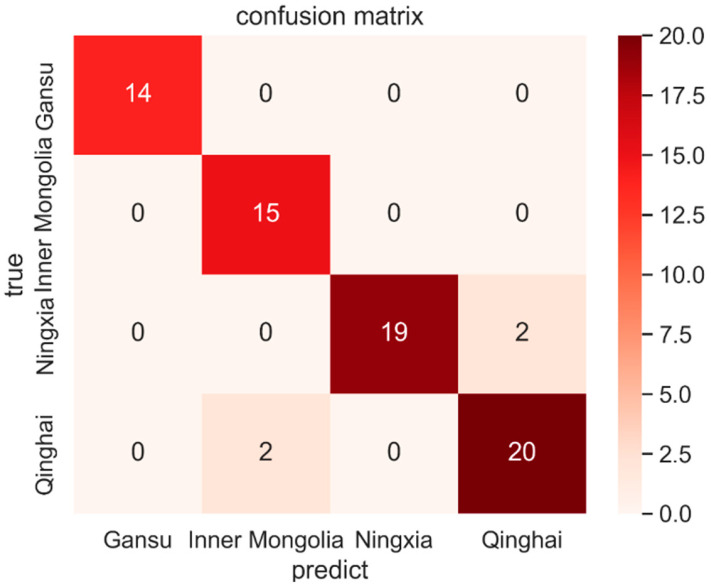
Confusion matrix heat map of one of the training results.

**Figure 12 foods-12-02541-f012:**
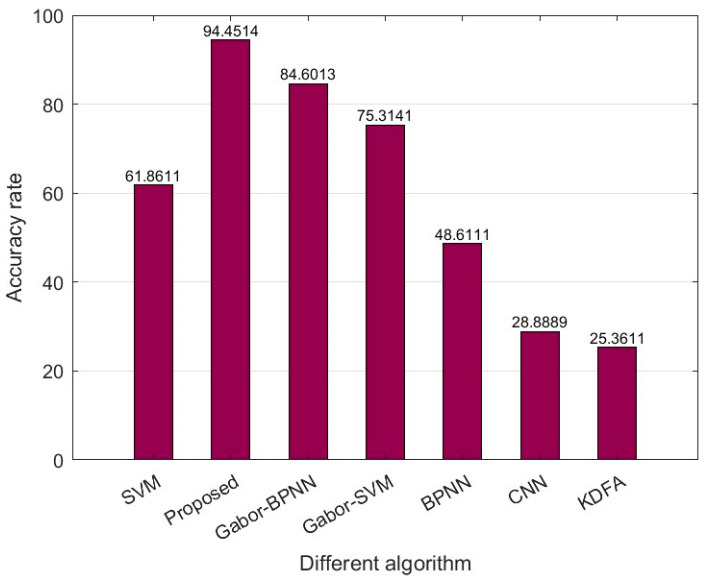
The average accuracy of the proposed method compared with six other methods.

**Figure 13 foods-12-02541-f013:**
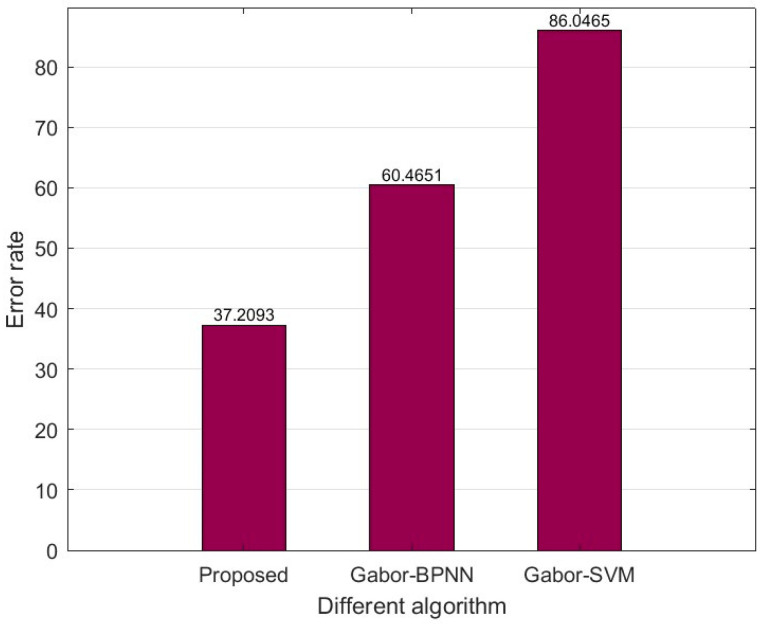
Proportion of Qinghai samples among samples with an error rate of more than 70%.

**Table 1 foods-12-02541-t001:** Wolfberry sample information sheet.

No.	Origin Type
GS.1-GS.90	Gansu
NM.1-NM.90	Inner Mongolia
NX.1-NX.90	Ningxia
QH.1-QH.90	Qinghai

**Table 2 foods-12-02541-t002:** Gabor transformation function parameter setting.

Channel	$\frac{1}{\lambda }$	θ	φ
H	13	$\frac{\pi }{4}$	0
S	10	$\frac{\pi }{3}$	0
I	10	$\frac{\pi }{12}$	0

**Table 3 foods-12-02541-t003:** Accuracy of the model under different hyperparameters.

nTree	mtry	Accuracy (%)
1000	50	93.33
2000	50	94.45
3000	50	92.22
2000	40	92.50
2000	60	90.28

**Table 4 foods-12-02541-t004:** Specific situation table of samples with an error rate higher than 90%.

Sample Name	Error Rate	Predicted Origin	Actual Origin
QH 5-1.JPG	100.00%	Ningxia	Qinghai
NX 16-1.JPG	100.00%	Qinghai	Ningxia
NX 13-2.JPG	100.00%	Qinghai	Ningxia
GS 42-2.JPG	100.00%	Ningxia	Gansu
NM 44-1.JPG	100.00%	Qinghai	Inner Mongolia
QH 8-1.JPG	100.00%	Inner Mongolia	Qinghai
QH 38-2.JPG	100.00%	Ningxia	Qinghai
QH 11-1.JPG	100.00%	Ningxia	Qinghai
NX 7-2.JPG	100.00%	Qinghai	Ningxia
QH 22-1.JPG	100.00%	Ningxia	Qinghai
QH 8-2.JPG	100.00%	Inner Mongolia	Qinghai
NX 44-1.JPG	100.00%	Qinghai	Ningxia
QH 20-2.JPG	100.00%	Ningxia	Qinghai
GS 12-2.JPG	100.00%	Qinghai	Gansu
QH 45-2.JPG	100.00%	Inner Mongolia	Qinghai
QH 17-1.JPG	100.00%	Ningxia	Qinghai
QH 13-2.JPG	100.00%	Ningxia	Qinghai
QH 26-2.JPG	100.00%	Inner Mongolia	Qinghai
NM 45-2.JPG	100.00%	Qinghai	Inner Mongolia
GS 38-1.JPG	100.00%	Ningxia	Gansu
NX 16-2.JPG	100.00%	Qinghai	Ningxia
GS 25-1.JPG	97.87%	Ningxia	Gansu
QH 38-1.JPG	97.73%	Inner Mongolia	Qinghai
GS 36-2.JPG	97.67%	Ningxia	Gansu
NM 32-1.JPG	97.37%	Gansu	Inner Mongolia
NM 36-1.JPG	96.30%	Qinghai	Inner Mongolia
QH 16-2.JPG	93.88%	Ningxia	Qinghai

## Data Availability

The data presented in this study are available on request from the corresponding author.

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
