# Peer review of "Intelligent Identification Method of Geographic Origin for Chinese Wolfberries Based on Color Space Transformation and Texture Morphological Features"

_foods, 2023, doi:10.3390/foods12132541_

Round 1
Reviewer 1 Report
This study is to distinguish the geographic origin of Chinese wolfberry using an intelligent identification method.
In Figure 1, Chinese wolfberry pictures by region in 'Image Preprocessing' are the same, so it is difficult to see any difference with the naked eye.
In Figure 5, these pictures are believed to be representative of the differences between Chinese wolfberry varieties by region.
The overall structure of the study is very good. However, the method introduced to distinguish berries is not correct. Image discrimination using machine learning is learning by looking at the appearance and finding feature points.
However, it is difficult to find features by machine learning for the image learned by Figure 5. In other words, even if you apply a very good image learning machine algorithm, the error will be high.
When judging by the characteristics of the picture in Figure 5, the appropriate method is proposed as follows.
1. Draw a line in the long direction of the berry.
2. Based on the line of 1, measure the length by dividing it into 5-6 equal parts at right angles.
3. Record the length of 5 to 6 equal parts of 2 as data.
4. If machine learning is performed on this data with the "Iris Dataset" classification method, it will be a successful classification.
None
Author Response
Response to Reviewer 1 Comments
Reviewer 1:
Thanks for the reviewer’s overall positive comments and further valuable constructive comments for improvement.
Point 1: This study is to distinguish the geographic origin of Chinese wolfberry using an intelligent identification method. In Figure 1, Chinese wolfberry pictures by region in 'Image Preprocessing' are the same, so it is difficult to see any difference with the naked eye. In Figure 5, these pictures are believed to be representative of the differences between Chinese wolfberry varieties by region. The overall structure of the study is very good. However, the method introduced to distinguish berries is not correct. Image discrimination using machine learning is learning by looking at the appearance and finding feature points. However, it is difficult to find features by machine learning for the image learned by Figure 5. In other words, even if you apply a very good image learning machine algorithm, the error will be high. When judging by the characteristics of the picture in Figure 5, the appropriate method is proposed as follows. 1. Draw a line in the long direction of the berry. 2. Based on the line of 1, measure the length by dividing it into 5-6 equal parts at right angles. 3. Record the length of 5 to 6 equal parts of 2 as data. 4. If machine learning is performed on this data with the "Iris Dataset" classification method, it will be a successful classification.
Response 1: Thank you for this advice. The texture of Chinese wolfberries from different origins are various [4]. Gabor transformation can extract the texture features in wolfberry images [20]. Figure 6 illustrates the images after Gabor transformation in I channel. From Figure 6 the texture of the wolfberry samples from different origins are various. Therefore, the histogram is calculated after Gabor transformation to reflects the distribution of different texture characteristics for the wolfberries from different origins. Considering the comment of the reviewer, we have added a discussion in the last paragraph of Section 2.3:" The texture of Chinese wolfberries from different origins are various [4]. Gabor transformation can extract the texture features in wolfberry images [20]. Figure 6 illustrates the images after Gabor transformation in I channel. From Figure 6 the texture of the wolfberry samples from Inner Mongolia and Qinghai is more rugged, with large ravine stripes. While the wolfberry samples from Gansu and Ningxia are more delicate. Therefore, the histogram is calculated after Gabor transformation to reflects the distribution of different texture characteristics for the wolfberries from different origins." We have also drawn the images after Gabor transformation in Fig. 6. We can see that texture of the wolfberry images from different origins are various.
Besides the Gabor transformation features, the Hu invariant moments are extracted to preserve the morphological characteristics of wolfberry. Figure 7 shows the Hu invariant moments of the wolfberry images from the four origins. It can be seen that there are differences in the Hu invariant moment characteristics of the four origins.
The experimental results show that the proposed method has a high accuracy rate. It proves that the features extracted in this paper are effective.
Thank you very much for your suggestion about the method. We will employ your suggestion in our future work about the recognition of the wolfberry origins.

Reviewer 2 Report
The paper presents a study on the feasibility of the identification of geographic origin of Chinese wolfberry using machine vision and machine learning. The paper needs major revisions since, in the present form, some parts are only superficially discussed. In particular:
1) The resolution of the initial images as well as the resolution of the images after pre-processing must be presented.
2) The sections of the image pre-processing, Gabor transformation and Hu invariant moment feature extraction are difficult to follow for the non-specialist since the methods are presented with a number of equations without any reference about where these equations come from. I think the authors should also discuss the methods in simple words and explain why and how these transformations helps in the geographical origin identification.
3) In my opinion the Section “Sensitivity to Training set scale” does not technically sound very much. The section discusses how increasing the ratio training samples to test samples the accuracy rate increases too. However, it is not clear if the accuracy rate is calculated on the training samples or the test samples. In the first case it is not correct to state that the accuracy rate is improving because this can be the result of data overfitting. If, instead, the accuracy rate is calculated on the test samples, it is not clear why in the following the authors present the results in the case of ratio 80-20 that is characterized by the minimum accuracy rate. I think it can be useful to plot the accuracy rate for both training samples and test samples and to choose the training-test ratio corresponding to the maximum accuracy rate with no overfitting.
4) Fig. 7 is of very low quality and must be improved.
5) I think the authors should include a confusion matrix to present the accuracy results.
6) The authors should proofread the manuscript to correct errors and typos. For example: in the abstract “the collect the image data” should be “to collect the image data”; page 2, instead of “high equipment” it is better “expensive instrumentation”; in first line of “Methods” “origin for Chinese wolfberry origin” should be “origin for Chinese wolfberry”; in equation 5 “G” must be subscript; in the second line of the section “Hu invariant moment feature extraction” “First convert the RGB images are converted” should be “First the RGB images are converted”; in the section “Model training” “testY be the label of training set” should be “testY be the label of test set”.
The authors must fix some errors and typos as discussed in the comments.
Author Response
Response to Reviewer 2 Comments
Reviewer 2:
Thanks for the reviewer’s overall positive comments and further valuable constructive comments for improvement.
Point 1: The resolution of the initial images as well as the resolution of the images after pre-processing must be presented.
Response 1: Yes, according to the advice of the reviewer, in the revised version of our manuscript we have added the resolution of the initial images in the third sentence of the last paragraph in Section 2.1:“ The pixels of the four pictures are 8688*5792.”. We have also added the resolution of the images after pre-processing in the 4th and 5th sentences of the last paragraph in Section 2.2:” The wolfberries in Figures 4(a), 4(b), 4(c), and 4(d) are from Gansu, Inner Mongolia, Ningxia and Qinghai, respectively. And the resolution of the pixels for the four images are 886*947, 853*573, 514*1249, and 533*1024, respectively.”.
Point 2: The sections of the image pre-processing, Gabor transformation and Hu invariant moment feature extraction are difficult to follow for the non-specialist since the methods are presented with a number of equations without any reference about where these equations come from. I think the authors should also discuss the methods in simple words and explain why and how these transformations helps in the geographical origin identification.
Response 2: Yes, thanks for this advice. The image pre-processing procedure is used to remove invalid background information. Gabor transformation is used to attract the texture information. And Hu invariant moment feature can preserve the morphological characteristics of wolfberry.
According to the advice of the reviewer, in the revised version of our manuscript we have added a discussion in the last paragraph of Section 2.2:” During the entire image preprocessing process, the above algorithms are used to complete the image cropping and background color unification. The purpose of this procedure is to reduce the interference of the background on subsequent model training. Figure 4 illustrates the preprocessed images of Chinese wolfberry from four origins. The wolfberries in Figures 4(a), 4(b), 4(c), and 4(d) are from Gansu, Inner Mongolia, Ningxia and Qinghai, respectively. And the resolution of the pixels for the four images are 886*947, 853*573, 514*1249, and 533*1024, respectively.” We have also drawn the images after image pre-processing in Fig. 4. We can see that the invalid background information in these images have been removed.
For Gabor transformation feature extraction, we have added a discussion in the last paragraph of Section 2.3: " The texture of Chinese wolfberries from different origins are various [4]. Gabor transformation can extract the texture features in wolfberry images [20]. Figure 6 illustrates the images after Gabor transformation in I channel. From Figure 6 the texture of the wolfberry samples from Inner Mongolia and Qinghai is more rugged, with large ravine stripes. While the wolfberry samples from Gansu and Ningxia are more delicate. Therefore, the histogram is calculated after Gabor transformation to reflects the distribution of different texture characteristics for the wolfberries from different origins." We have also drawn the images after Gabor transformation in Fig. 6. We can see that texture of the wolfberry images from different origins are various.
As for Hu invariant moment feature extraction, we have added the discussion in the last paragraph of Section 2.4: " According to the above formulas, the Hu invariant moments of the wolfberry images are calculated. Figure 7 shows the Hu invariant moments of the wolfberry images from the four origins. It can be seen that there are differences in the Hu invariant moment characteristics of the four origins." We have also illustrated the Hu invariant moment information in Fig. 7. We can see that features can distinguish the wolfberries from different origins.
Point 3: In my opinion the Section “Sensitivity to Training set scale” does not technically sound very much. The section discusses how increasing the ratio training samples to test samples the accuracy rate increases too. However, it is not clear if the accuracy rate is calculated on the training samples or the test samples. In the first case it is not correct to state that the accuracy rate is improving because this can be the result of data overfitting. If, instead, the accuracy rate is calculated on the test samples, it is not clear why in the following the authors present the results in the case of ratio 80-20 that is characterized by the minimum accuracy rate. I think it can be useful to plot the accuracy rate for both training samples and test samples and to choose the training-test ratio corresponding to the maximum accuracy rate with no overfitting.
Response 3: Yes, thanks for pointing out this. Firstly, the purpose of this analysis is to prove the stability of the model under different proportions of the training set. Considering the reviewer's suggestion, in our revised version of the manuscript, we have added the purpose of the Section “Sensitivity to Training set scale” in the 1st-3rd sentences of Section 3.2: "In the proposed method, the training set samples are trained to obtain the prediction model. Thus, the training set size can affect prediction accuracy. In order to analyze the stability of the proposed method, 9 groups of comparative experiments are set up.". Secondly, the accuracy rate is calculated on the test samples. Thirdly, the ratio 80-20 is the commonly used ratio in machine learning. This ratio means that the training set occupies 80% and the test set occupies 20%. Because this ratio is very classic, it is used in the Sections “Determination of the hyperparameters of RF”, “Misjudgement analysis” and “Performance comparison”. Finally, all the results in Fig. 8 are averages obtained from 200 rounds, which is fair.
Point 4: Fig. 7 is of very low quality and must be improved.
Response 4: Yes, thanks for this advice. Fig. 7 in the previous version of the manuscript is Fig. 9 in the revised version of the manuscript. According to the advice of the reviewer, in the revised version of our manuscript we have redrawn this figure and improved its quality.
Point 5: I think the authors should include a confusion matrix to present the accuracy results.
Response 5: Yes, thanks for this advice. According to the advice of the reviewer, in the revised version of our manuscript we have added a confusion matrix to present the accuracy results in Fig. 11. In addition, we have added the analysis of the figure in the last paragraph of Section 3.3: ”Figure 11 illustrates the heat map of the confusion matrix obtained from one of the training results. It can be seen that while the model proposed in this paper has a high accuracy rate, Ningxia wolfberry is easily misjudged as Qinghai wolfberry.”.
Point 6: The authors should proofread the manuscript to correct errors and typos. For example: in the abstract “the collect the image data” should be “to collect the image data”; page 2, instead of “high equipment” it is better “expensive instrumentation”; in first line of “Methods” “origin for Chinese wolfberry origin” should be “origin for Chinese wolfberry”; in equation 5 “G” must be subscript; in the second line of the section “Hu invariant moment feature extraction” “First convert the RGB images are converted” should be “First the RGB images are converted”; in the section “Model training” “testY be the label of training set” should be “testY be the label of test set”.
Response 6: Yes, thanks for pointing out this. According to the advice of the reviewer, we have carefully checked our manuscript and revised errors and typos. For example: in the abstract “the collect the image data” has been corrected to be “to collect the image data”. In the last sentence of the second paragraph in the last sentence of the second paragraph in Section 1, “high equipment” has been revised to be “expensive instrumentation”. In the first sentence of the first paragraph in Section 2, “origin for Chinese wolfberry origin” has been corrected to be “origin for Chinese wolfberry”. In equation 5, “G” has been revised to be subscript. In the second sentence of the first paragraph in Section 2.4, “First convert the RGB images are converted” has been corrected to be “First the RGB images are converted”. In the third sentence of the first paragraph in Section 2.5, “testY be the label of training set” has been corrected to be “testY be the label of test set”.

Reviewer 3 Report
In this study, the authors proposed a method to predict the origin information of a single wolfberry image using color space transformation and texture morphological features. The obtained accuracies were high and the manuscript is interesting, well-written and easy to read.
My main critical point is about the lack of information about the machine learning algorithms.
Why didn't you use the template?
Introduction
It was unclear the reasons why the random forests were chosen.
Model training
How did you optimize the hyperparameters of RF?
You evaluated the sensitivity to Training set scale, however, didn't analyze the importance of variables. Please clarify which metrics were effective.
There is no discussion.
Author Response
Response to Reviewer 3 Comments
Reviewer 3:
Thanks for the reviewer’s overall positive comments and further valuable constructive comments for improvement.
Point 1: The lack of information about the machine learning algorithms.
Response 1: Yes, thanks for pointing out this. Following the reviewer's suggestion, in the revised version of our manuscript we added the information about the machine learning algorithms in the 2nd to 5th sentences of fourth paragraph in Section 1: "Machine learning is a branch of artificial intelligence, which enables computers to automatically learn the laws of data through models and algorithms, so as to realize the prediction and classification of new data. There are mainly supervised, unsupervised, semi-supervised and re enforcement learning types, among which supervised learning is the most common. Algorithms are very important in machine learning, such as linear regression, decision trees, neural networks, etc., and feature engineering is also an important link. Machine learning technology is widely used in natural language processing, image recognition, medical diagnosis and other fields." In addition, the random forest information has been added in the 1st and 2nd sentences of the second paragraph in Section 2.5:” The random forest algorithm is an integrated learning algorithm based on decision trees, which performs classification and regression by constructing multiple decision trees. Each decision tree in random forest is built by randomly selecting samples and features, so it has good generalization ability.”.
Point 2: Why didn't you use the template?
Response 2: Yes, thanks for this advice. According to the advice of the reviewer, in the revised version of our manuscript we have used the template.
Point 3: It was unclear the reasons why the random forests were chosen.
Response 3: Yes, thanks for pointing out this. The random forest algorithm is an integrated learning algorithm based on decision trees, which performs classification and regression by constructing multiple decision trees. Each decision tree in random forest is built by randomly selecting samples and features, so it has good generalization ability. In the performance comparison part of Section 3.4, compared with SVM and BPNN algorithms, it is found that the average accuracy of the random forest algorithm is the highest, so the random forest algorithm is adopted.
Point 4: How did you optimize the hyperparameters of RF?
Response 4: Yes, thanks for this advice. We choose the hyperparameters of RF with the highest accuracy result. Considering the suggestion of the reviewer, we have added Section 3.1 to show the determination of the hyperparameters of RF: “In order to determine the optimize hyperparameters of RF, 5 groups of comparative experiments were executed. The ratio of training set to test set in each experiment is 8:2. The hyperparameter settings of the RF model are shown in Table 3. It can be concluded that when nTree is set to 2000 and mtry is set to 50, the model works best. Thus, in the prediction model nTree and mtry are set as 2000 and 50, respectively.”.
Piont 5: You evaluated the sensitivity to Training set scale, however, didn't analyze the importance of variables. Please clarify which metrics were effective.
Response 5: Yes, thanks for pointing out this. nTree and mtry are important variables in the proposed method. According to the advice of the reviewer, in the revised version of our manuscript we have added a Section 3.1 to show the determination of the hyperparameters of RF: "In order to determine the optimize hyperparameters of RF, 5 groups of comparative experiments were executed. The ratio of training set to test set in each experiment is 8:2. The hyperparameter settings of the RF model are shown in Table 3. It can be concluded that when nTree is set to 2000 and mtry is set to 50, the model works best. Thus, in the prediction model nTree and mtry are set as 2000 and 50, respectively.".

Round 2
Reviewer 2 Report
The paper has been revised according the the Reviewer comments. It can be accepted for publication.
Reviewer 3 Report
The authors added some elements to enrich manuscript and I think this paper can now be accepted for publication.